# Muscular and Kinematic Responses to Unexpected Translational Balance Perturbation: A Pilot Study in Healthy Young Adults

**DOI:** 10.3390/bioengineering10070831

**Published:** 2023-07-13

**Authors:** Cheuk Ying Tong, Ringo Tang-Long Zhu, Yan To Ling, Eduardo Mendonça Scheeren, Freddy Man Hin Lam, Hong Fu, Christina Zong-Hao Ma

**Affiliations:** 1Department of Biomedical Engineering, The Hong Kong Polytechnic University, Hong Kong SAR 999077, China; cheuk-ying.tong@connect.polyu.hk (C.Y.T.); ringo-tanglong.zhu@connect.polyu.hk (R.T.-L.Z.); jane.yt.ling@connect.polyu.hk (Y.T.L.); 2Research Institute for Smart Ageing, The Hong Kong Polytechnic University, Hong Kong SAR 999077, China; 3Centre for Developmental Neurobiology, King’s College London, London SE1 1UL, UK; 4Graduate Program in Health Technology, Pontifícia Universidade Católica do Paraná, Curitiba 80215-901, Brazil; eduardo.scheeren@pucpr.br; 5Department of Rehabilitation Sciences, The Hong Kong Polytechnic University, Hong Kong SAR 999077, China; freddy-mh.lam@polyu.edu.hk; 6Department of Mathematics and Information Technology, The Education University of Hong Kong, Hong Kong SAR 999077, China

**Keywords:** translational balance perturbation, moving platform, muscle activation, muscle co-contraction, onset latency, time to peak, electromyography (EMG), mechanomyography (MMG)

## Abstract

Falls and fall-related injuries are significant public health problems in older adults. While balance-controlling strategies have been extensively researched, there is still a lack of understanding regarding how fast the lower-limb muscles contract and coordinate in response to a sudden loss of standing balance. Therefore, this pilot study aims to investigate the speed and timing patterns of multiple joint/muscles’ activities among the different challenges in standing balance. Twelve healthy young subjects were recruited, and they received unexpected translational balance perturbations with randomized intensities and directions. Electromyographical (EMG) and mechanomyographical (MMG) signals of eight dominant-leg’s muscles, dominant-leg’s three-dimensional (3D) hip/knee/ankle joint angles, and 3D postural sways were concurrently collected. Two-way ANOVAs were used to examine the difference in timing and speed of the collected signals among muscles/joint motions and among perturbation intensities. This study has found that (1) agonist muscles resisting the induced postural sway tended to activate more rapidly than the antagonist muscles, and ankle muscles contributed the most with the fastest rate of response; (2) voluntary corrective lower-limb joint motions and postural sways could occur as early as the perturbation-induced passive ones; (3) muscles reacted more rapidly under a larger perturbation intensity, while the joint motions or postural sways did not. These findings expand the current knowledge on standing-balance-controlling mechanisms and may potentially provide more insights for developing future fall-prevention strategies in daily life.

## 1. Introduction

Falls are one of the major public health problems in the world. Approximately one in three older adults fall worldwide [1], and 28.7% of older adults fall in the United States annually [2]. Every year, there are around 684,000 fatal falls, and it is the second leading cause of unintentional injury death [1,3]. For the non-fatal injuries, about one in ten older adults experiences a fall-related injury annually [4]. Falls cause physical and mental impacts on older adults and can further heavily burden society [1]. Balance and gait disorder is the major cause of falls excepting accidents [5]. The appropriate and timely postural adjustments and lower-limb muscle activities are vital to maintain postural balance. The in-depth investigation into these strategies for how a person reacts to sudden balance perturbations can facilitate our understanding of the underlying mechanisms of falls. This could also provide more insights and inspire the future development of fall-prevention strategies.

Humans have different patterns of lower-limb joint motions in response to varied intensities of balance perturbations. Previous studies have identified three fundamental strategies to maintain the static standing balance, i.e., the ankle strategy, hip strategy, and stepping strategy [6]. The ankle strategy is reached mainly by the dorsiflexion and plantarflexion of ankle joint, with minimal movement of the other proximal joints [7]. This strategy is dominantly employed when no external perturbation exists [8]. The hip strategy is used when the ankle strategy is not enough to keep the center of mass (COM) within the base of support (BOS) [7]. If the ankle and hip muscles cannot contract sufficiently to compensate for the large balance perturbations, the stepping strategy would be employed [9]. Sometimes a “mixed strategy” can also be used to maintain balance with typical characteristics of using the combined above-mentioned strategies depending on the specific situation [10].

An investigation of the rapidity and appropriate sequence of movements in the hip, knee, and ankle joints is crucial for understanding the balance-controlling strategy, especially during unexpected and intense balance perturbations. Previous studies have primarily focused on the peak responses of postural sways and joint angles in maintaining standing balance [11,12,13], and only a few studies have analyzed the onset sequence of various lower-limb joint motions in response to balance perturbations in the sagittal plane [14]. When specifically looking at the lower-limb responses following the translational perturbations induced by a suddenly forward-moving platform, the lower-limb joints commonly react with the sequence of ankle dorsiflexion, knee flexion, and finally hip flexion [14]. However, research on the timing and speed of lower-limb joint responses to balance perturbations in the frontal plane remains inadequate. Therefore, a study of the temporal parameters, such as onset latency and time to peak, of whole-body postural sways as well as lower-limb joint motions in response to the balance perturbations with various intensities and directions is warranted.

In addition to examining the lower-limb joint motions, analyzing the lower-limb muscle activities can provide greater insights into the underlying balance-controlling strategies. Irrespective of the balance-controlling strategy utilized, the acceleration of any body segment resulting from a perturbation must be generated by the contraction of the corresponding skeletal muscles. Most previous studies investigated the signals of only one or a couple of lower-limb muscles for maintaining balance [15,16,17,18,19,20,21]. Specifically, most previous studies on static balance control mainly investigated the EMG signals of the ankle dorsiflexor and plantarflexor [6,15,16,17]. During the walking task with unexpectedly induced slipping, the previous literature has reported that older adults who failed to maintain balance tended to have delayed EMG onset in the knee flexors/extensors of the slipping legs [18,19]. Some other studies also reported that the large rate of EMG rise in the ankle plantarflexor and knee flexor of the stance leg was important for preventing tripping [20] and that in the hip abductors/adductors was important for protective stepping [21] in older people. However, until now, there has been insufficient evidence on how fast the major hip, knee, and ankle muscles can react to the balance perturbations with varying intensities and directions.

Meanwhile, one of our recent works investigated the rapid responses of eight dominant-leg muscles following the sudden loss of balance induced by waist-pull perturbations, by quantifying the EMG-onset latency, time to peak EMG amplitude, and rate of EMG rise of the captioned muscles [22]. It mainly identified that the agonist muscles exhibited quicker activation than the antagonist muscles, and ankle muscles tended to activate faster than the remaining six muscles (i.e., hip flexor/extensor, hip abductor/adductor, and knee flexor/extensor) in response to the waist-pull balance perturbation in young adults [22]. However, it has remained unclear whether young adults would adopt similar strategies in response to the moving-platform balance perturbation, which mimics a more real-life situation of standing in the buses/trains/boats and merits further study.

Apart from the joint reactions and muscle electrical activities, the response of other events along the motor output pathway during balance control has been less explored. The generation of a joint motion goes through activation in neuromuscular junctions, muscle mechanical activities (shortening and lateral vibrations), and force propagation to tendons [23]. However, it is unknown whether the timing of muscle mechanical activities plays a role in the recovery from balance perturbations of varying intensities and directions. Mechanomyography (MMG) can detect such mechanical activities of a contracting muscle by recording the lateral vibrations that are perpendicular to the muscle fiber direction on the skin surface [24]. Previous research has reported that the onset of the MMG signal was later than that of the EMG signal during the isometric contraction [24]. For dynamic situations, MMG signals have been investigated in the tasks of maintaining walking balance [25,26] and standing balance [22]. The MMG peak timing was found to be later than the EMG peak timing for the ankle plantarflexor during a gait cycle [25]; however, the onset and peak timing of MMG signals were found generally earlier than that of EMG signals following the sudden perturbations induced by waist pulls, which might be affected by the noise of passive body-segment movements induced by perturbations [22]. Therefore, this study made further attempts to explore the timing and coordination patterns of major lower-limb muscles’ mechanical activities during balance control.

To bridge the above research gaps, this study aimed to explore how healthy young adults respond to the moving-platform-induced balance perturbations with multiple directions and intensities from the perspectives of postural sways, lower-limb joint motions, and lower-limb muscle activities. The objectives of this study were to examine the differences in speed (onset latency, time to peak, and/or rate of rise) and peak responses of (1) the forward/backward, medial/lateral, and upward/downward COM displacements; (2) the eight lower-limb joint motions (i.e., hip flexion/extension, hip abduction/adduction, knee flexion/extension, and ankle dorsiflexion/plantarflexion); and (3) the eight dominant-leg muscles’ electrical and mechanical activities (i.e., hip flexor/extensor, hip abductor/adductor, knee flexor/extensor, and ankle dorsiflexor/plantarflexor) under the different perturbation intensities. Such comprehensive analysis of kinematics and muscle activities under the sudden loss of balance that simulates daily scenarios is expected to uncover the more in-depth mechanisms underlying standing-balance control. We hypothesized that (1) agonist muscles resisting the induced postural sway would have faster activation than antagonist muscles; (2) involuntary/passive joint motions and postural sways induced by the unexpected perturbation would occur earlier than voluntary/active ones for balance recovery; and (3) higher intensities of unexpected perturbations would induce faster muscle activities, joint motions, and postural sways.

## 2. Materials and Methods

### 2.1. Study Design and Subjects

This is an observational/cross-sectional and exploratory study. A total of 12 young healthy adults aged between 18 to 24 were recruited through the method of convenience sampling. Ethical approval was granted by the Institutional Review Board (IRB) of The Hong Kong Polytechnic University (HSEARS20201230002). The subjects satisfied the following inclusion criteria [27]:No high-intensity sports within 24 h before the experiment;No known neurological or musculoskeletal deficits;No history of balance disorders, walking disorders, or dizziness;No history of lower-limb injuries within a week;No sight or hearing disorders;No medication intake that could affect muscle activities.

### 2.2. Equipment

#### 2.2.1. Moving-Platform Perturbation System

A moving-platform perturbation system was developed to induce the unexpected balance perturbations (see Figure 1). It consisted of (1) an aluminum alloy frame, (2) four servo motors (130-07725AS4, Wenzhou Guomai Electronics Ltd., Wenzhou, China), (3) a customized circular wooden platform (diameter: 80.0 cm; thickness: 3.5 cm), (4) four braided polyethylene wires (diameter: 1.2 mm), (5) a set of rails, and (6) a safety harness system (PG-360, Physio Gait Dynamic Unweighting System, Healthcare International Ltd., Langley, WA, USA). An Arduino UNO board (Arduino Uno Rev3, The Arduino Team, Somerville, America) and a customized Arduino program were used to control the servo motors and deliver the unexpected balance perturbations via the wires. Each of the four wires connected the edge of the platform with a motor, and the motor would pull the platform to slide along the rails so as to induce the horizontal balance perturbation in one of the four directions with regard to the subject’s body (anterior, posterior, left, and right). Figure 2 shows the flow of generating one perturbation. Firstly, the system delivered a sudden pull to the wooden platform with a randomized direction and intensity. Then, the platform would remain stationary for 8 s. Finally, the system pulled the wooden platform to return to its original position. The direction, intensity, and starting time of each perturbation were randomized.

#### 2.2.2. Data Sampling Equipment

A three-dimensional capture and analysis system, i.e., the Vicon system (Nexus 2.11, Vicon Motion Systems Ltd., Yarnton, UK), with eight cameras, was applied to capture the whole-body kinematics. The sampling rate was 250 Hz. A total of 39 reflective markers were adhered to the subject’s body based on the full-body plug-in-gait dynamic model, including twelve on the bilateral lower limbs (bilateral thigh, lateral condyle of femur, shank, lateral malleolus, heel, and 2nd metatarsal head), four on the pelvis (bilateral anterior superior iliac spine [ASIS], and bilateral posterior superior iliac spine), five on the torso (sternal notch, xiphoid process of the sternum, spinous process of the 7th cervical vertebra, spinous process of 10th thoracic vertebra, and right scapula), fourteen on the bilateral upper limbs (bilateral acromion, upper arm, lateral epicondyle of humerus, forearm, radial styloid process, ulnar styloid process, and 3rd metacarpal head), and four on the head (bilateral temples and bilateral back head) [28]. To ensure the reflective markers were firmly fixed on the subject’s body surface, tight clothes and shorts were provided for subjects to wear during the experiment.

Eight major lower-limb muscles’ activities were collected by an eight-channel Trigno Wireless Biofeedback System (SP-W02D-1110, Delsys Inc., Natick, MA, USA). Each Delsys Trigno Avanti sensor (mass: 14 g; dimension: 37 mm × 27 mm × 13 mm) consisted of an EMG sensor (double-differential silver bar electrodes; inter-electrode distance: 10 mm; each electrode size: 5 mm × 1 mm; analogue Butterworth filter bandwidth: 20–450 Hz) and a 3-axis accelerometer (range: ±16 g; resolution: 10 bits) to serve as the MMG sensor [29]. The sampling rates of the EMG and MMG were 2000 Hz and 250 Hz, respectively. According to the Surface ElectroMyoGraphy for the Non-Invasive Assessment of Muscles (SENIAM) recommendation [30], the subject’s skin was shaved to remove the hair, and alcohol wipes were used to clean the skin surface. After the alcohol vaporized and when the skin was dry, the eight EMG and MMG sensors were placed over the investigated eight lower-limb muscles, respectively (see Table 1), and firmly fixed using double-sided adhesives (Adhesive Interfaces for Trigno Sensors, Delsys, Boston, MA, USA) and pressure-sensitive tapes (Haishi Hainuo Group, Qingdao, China). The Trigno Wireless Biofeedback System was commercially synchronized with the Vicon system. Infrared light bulbs were connected to the moving-platform perturbation system, and the flash of infrared light indicating each perturbation could be detected by the Vicon system. In this way, the three systems were synchronized during data collection.

### 2.3. Protocol

#### 2.3.1. Subjective Assessment

Each subject firstly accomplished the demographic data collection and the subjective assessments. Body mass and height were measured using a standard scale (DETECTO 3P704, Webb City, MI, USA), and other anthropometrics such as leg length were measured using a tape measure and a caliper. The International Physical Activity Questionnaire—Short version (IPAQ-S) [33] and the Falls Efficacy Scale—International (FES-I) short version [22,34,35,36] were used to evaluate the physical and mental factors that might affect the balance performance in each subject, respectively. A larger value calculated from the IPAQ-S reflects a higher level of physical activity in the past 1 week, and a higher score of the FES-I short version (7 items; full score: 28 points) reflects greater concerns over falling [33,34]. The Mini-Balance Evaluation System Test (Mini-BEST) was also performed to evaluate the subject’s balance capacity in four categories: anticipatory postural control, reactive postural control, sensory organization, and dynamic gait [37]. A higher score of the Mini-BEST (14 items; full score: 28 points) indicates better balance capacity. Finally, the subject’s dominant leg was ascertained using the balance recovery test, ball-kick test, and step-up test [38]. The leg used most frequently in nine total trials (3 trials × 3 tests) was considered the dominant leg [38].

#### 2.3.2. Instrumented Data Collection

Before the perturbation experiment, the reflective markers and the EMG and MMG sensors were attached to subjects, and instructions and explanation of the experimental protocol were given. To simulate the daily situation, the subject was asked to wear his/her usual shoes during the whole perturbation experiment. The subject was instructed to stand naturally with two feet shoulder-width apart in the middle of the platform and knees fully extended; subjects were instructed that when the platform moved, he/she should try his/her best to maintain balance without stepping or return to the original foot position as soon as possible if stepped. The subject held a light bar in front of the body at the waist level, and the reflective markers was not blocked during data collection in this way [11]. Dark-colored tapes were adhered below the subject’s shoes to mark the original foot position on the wooden platform.

Each subject received a total of 48 unexpected balance perturbations induced by the moving platform (4 directions × 4 intensities × 3 repetitions), and the kinematic, EMG, and MMG data of the subject’s responses were collected. These balance perturbations were randomly allocated into four perturbation trials during the experiment, and there was a 5 min rest between the two trials to avoid the effects of fatigue. For each perturbation, the starting time, direction (anterior, posterior, medial, or lateral), or intensity (highest, high, low, or lowest) was random. The direction of a balance perturbation was defined as the moving direction of the platform in reference to the subject’s dominant leg. For example, for a subject with the right leg as the dominant leg, pulling the platform toward the left was regarded as a “medial” perturbation, while pulling toward the right was a “lateral” perturbation (see Figure 1). Different intensities of balance perturbations were induced by the different displacements and velocities of the moving platform. Based on previous works and our pilot studies, the displacements under the “highest” intensity for the perturbations in anterior, posterior, medial, and lateral directions were set as 4%, 2.67%, 5.33%, and 5.33% of the subject’s height, respectively [22,39,40,41]. The displacements under the “high”, “low”, and “lowest” intensities corresponded to 3/4, 2/4, and 1/4 of that under the “highest” intensity, respectively. The pulling duration of each perturbation was measured from the flash time of the infrared light (see Appendix A). The displacement, velocity, and acceleration of each perturbation were measured based on the trajectory of the reflective marker fixed on the moving platform (see Appendix A). By examining these parameters, the moving-platform perturbation system showed good reliability in delivering three repetitive balance perturbations with the same direction and intensity (see Appendix A). Videos were taken during the entire perturbation experiment to evaluate the subject’s stepping strategy following the unexpected perturbations.

### 2.4. Data Processing

The kinematic data, i.e., the whole body’s center of mass (COM) and the hip, knee, and ankle angles, were firstly processed by the plug-in-gait dynamic model of the Vicon system and then zeroed to the baseline, i.e., mean signal value over the 1000 ms interval before each perturbation via a customized MATLAB program (MATLAB 2019b, The MathWorks, Inc., Natick, MA, USA). The COM displacement was further subtracted from the displacement of the platform to obtain the COM displacement relative to the base of support (BOS).

For the muscle activity data, the raw EMG data were firstly zeroed to the mean signal value over the whole perturbation trial, then full-wave rectified, and low-pass filtered by a bi-directional 4th-order Butterworth filter (cut-off frequency: 4 Hz) to obtain the EMG signal envelopes [22]. To extract MMG data, the z-axis accelerometry signals were firstly filtered through an adaptive filter in an attempt to eliminate the noise caused by limb motion by removing the trajectory of the reflective marker that was close to the MMG sensor. Next, the signals were further band-pass filtered by a 4th-order Butterworth filter (5–50 Hz) [25,26], full-wave rectified, and low-pass filtered by a bi-directional 4th-order Butterworth filter (cut-off frequency: 4 Hz) to obtain the MMG signal envelopes [42]. The EMG and MMG signal envelopes were further divided by the 1000 ms baseline mean values before the whole perturbation trial for normalization.

The onset latency, time to peak amplitude, peak amplitude, and/or rate of rise were analyzed for the kinematic, EMG, and MMG data (see Figure 3). The onset and peak points were identified within 2 s after the start of each perturbation. The onset point of a signal was determined as the first time point when the corresponding amplitude exceeded five times of standard deviation (SD) plus the mean of baseline (mean + 5 SD) [43,44]. The mean of baseline was calculated from the 1000 ms signal values before the start of each perturbation. The onset latency indicated the delayed time between the start of each perturbation and the onset point of a signal. The time to peak indicated the delayed time between the start of each perturbation and the peak point of a signal. The rate of rise indicated the slope of signal rise over the 50 ms interval after onset. For each parameter (i.e., onset latency, time to peak, peak value, or rate of rise), each subject’s mean of the three values in three repetitive perturbation trials with the same direction and same intensity was used for the statistical analysis.

### 2.5. Statistical Analyses

The IBM SPSS version 25 was used for statistical analyses, and the significance level was set as 0.05. For each of the four perturbation directions, two-way analysis of variance and post hoc pairwise comparisons with Bonferroni corrections were applied to analyze the effects of the below two factors on the onset latency, time to peak, peak amplitude, or rate of rise of the investigated signals:EMG signal difference among the eight different muscles and among the four different perturbation intensities (muscle × perturbation intensity);MMG signal difference among the eight different muscles and among the four different perturbation intensities (muscle × perturbation intensity);Joint angle difference among the eight different joint motions and among the four different perturbation intensities (joint motion × perturbation intensity);COM trajectory difference among the six different postural sway directions and among the four different perturbation intensities (postural sway direction × perturbation intensity).

## 3. Results

The demographic data and the subjective assessment results of 12 subjects are shown in Table 2. All subjects had the right leg as the dominant leg. No fall or disastrous event took place during experiments. All subjects reported that the safety harness system provided adequate protection while not restricting their motions.

Under the lowest, low, and high perturbation intensities, all young subjects were able to maintain balance without stepping or lifting their feet (0/432). Under the highest intensity of perturbations, eight subjects stepped for a total of 14 times following the anterior perturbations (14/36), no subject stepped following the posterior perturbation (0/36), one subject took a step with the non-dominant leg following the medial perturbation (1/36), and one subject took a step with the dominant leg following the lateral perturbation (1/36). Among the stepping responses following anterior perturbations, six subjects took two steps to maintain balance (7/14), three took one step with the dominant leg (4/14), and three took one step with the non-dominant leg (3/14).

### 3.1. COM Displacements

Figure 4 plots the mean whole-body COM displacement relative to the BOS of twelve subjects following unexpected balance perturbations in each of the four directions and four intensities (*n* = 12). A red dotted line specifies the start of the balance perturbation (t = 1 s). The mean and standard error (mean ± SE, *n* = 12) values of onset latencies, time to peak, and peak values are also plotted against the different directions of COM displacement in Figure 5.

For each of the four perturbation directions, the initial COM displacement was found in the direction opposite to the perturbation, followed by the COM displacement toward the perturbation significantly (*p* < 0.05). Perturbation intensity showed significant effect on the peak COM displacement (*p* < 0.05) but not on the timing of COM displacement.

The unexpected perturbation also induced the early response of COM displacement in the vertical direction. Following anterior perturbations, the COM displacement had significantly earlier onset and significantly shorter time to peak in the upward direction than in downward/forward/medial/lateral directions (*p* < 0.05). Following posterior perturbations, the significantly shorter time to peak COM displacement was observed in the downward direction compared to the upward/backward/medial/lateral directions (*p* < 0.05). Following medial perturbations, the COM displacement in the upward direction had significantly earlier onset and significantly shorter time to peak than in downward/forward/backward direction (*p* < 0.05). Following lateral perturbations, the COM displacement had significantly earlier onset and significantly shorter time to peak in the upward direction than in downward/lateral/forward/backward direction (*p* < 0.05).

### 3.2. Dominant-Leg Joint Motions

Figure 6 plots the mean dominant-leg joint angle changes of the twelve subjects following unexpected translational balance perturbations (*n* = 12). Figure 7 shows the mean and standard error (mean ± SE, *n* = 12) values of onset latencies, time to peak, and peak values of the dominant-leg joint angles.

When anterior perturbation was induced, significant differences within joints existed in the angle-onset latencies (hip flexion < hip extension; hip adduction < hip abduction; *p* < 0.05) and in the time to peak angles (hip flexion < hip extension; hip adduction < hip abduction; knee flexion < knee extension; *p* < 0.05). Generally, the peak angles of hip flexion, knee flexion, and ankle dorsiflexion significantly increased with the perturbation intensity (*p* < 0.05).

When posterior perturbation was induced, ankle dorsiflexion had significantly earlier angle onset and significantly shorter time to peak angle than ankle plantarflexion (*p* < 0.05). Among the eight joint motions, the significantly largest joint motion occurred in hip flexion (*p* < 0.05). Generally, larger perturbation intensity induced larger dominant-leg joint motions (*p* < 0.05).

When medial perturbation was induced, hip adduction showed the significantly earliest angle onset among the eight joint motions (*p* < 0.05). Hip adduction also showed significantly shorter time to reach peak angle than hip abduction under the low, high, and highest perturbation intensities (*p* < 0.05), while ankle dorsiflexion showed significantly shorter time to reach peak angle than ankle plantarflexion under the highest perturbation intensity (*p* < 0.05). Generally, the peak angles of hip flexion, knee flexion, and ankle dorsiflexion significantly increased with the perturbation intensity (*p* < 0.05).

When lateral perturbation was induced, significant differences within joints existed in both the angle-onset latency and the time to peak angle (hip abduction < hip adduction; hip flexion < hip extension; knee flexion < knee extension; ankle dorsiflexion < ankle plantarflexion; *p* < 0.05). Under the low, high, and highest perturbation intensities, hip flexion and knee flexion had significantly larger peak angles than the rest of the joint motions (*p* < 0.05).

### 3.3. EMG Signals of Eight Dominant-Leg Muscles

Figure 8 shows the mean EMG signal of the twelve subjects for each of the eight dominant-leg muscles following the unexpected balance perturbations (*n* = 12). Figure 9 plots the mean and standard error (mean ± SE, *n* = 12) values of EMG-onset latencies, time to peak EMG amplitude, as well as rate of EMG rise against the eight dominant-leg muscles. 

Following sudden anterior movement of the platform, significant differences within the agonist–antagonist muscle pairs were found in EMG-onset latency (ankle dorsiflexor < ankle plantarflexor for the lowest, low, and high perturbation intensities; hip flexor < hip extensor for the lowest perturbation intensity; *p* < 0.05) and in time to peak EMG amplitude (ankle dorsiflexor < ankle plantarflexor for all perturbation intensities; knee extensor < knee flexor for the lowest perturbation intensity; *p* < 0.05). The rate of EMG rise was remarkably highest for the ankle dorsiflexor compared to the other muscles (*p* < 0.05).

Following sudden posterior movement of the platform, the ankle plantarflexor had a significantly shorter EMG-onset latency compared to the knee extensor, hip extensor, and hip adductor (*p* < 0.05). Among the eight dominant-leg muscles, the ankle plantarflexor had the significantly largest rate of EMG rise (*p* < 0.05). Generally, larger perturbation intensities induced significantly shorter EMG-onset latencies (*p* < 0.05) and a significantly larger rate of EMG rise (*p* < 0.05).

Following sudden medial movement of the platform, significant differences within the agonist–antagonist muscle pairs were found in EMG-onset latency (hip abductor < hip adductor; *p* < 0.05) and time to EMG peak amplitude (hip abductor < hip adductor; knee extensor < knee flexor; *p* < 0.05). The ankle dorsiflexor had generally the largest rate of EMG rise among the eight dominant-leg muscles under the low, high, and highest perturbation intensities (*p* < 0.05). Generally, larger perturbation intensities induced significantly shorter EMG-onset latencies (*p* < 0.05) and significantly shorter time to EMG amplitude (*p* < 0.05); for the ankle dorsiflexor, ankle plantarflexor, and hip abductor, the rate of EMG rise also significantly increased with the perturbation intensity (*p* < 0.05).

Following sudden lateral movement of the platform, significant agonist–antagonist differences were found in EMG-onset latency (ankle dorsiflexor < ankle plantarflexor; knee extensor < knee flexor; hip flexor < hip extensor; *p* < 0.05) and in time to peak EMG amplitude (ankle dorsiflexor < ankle plantarflexor; hip adductor < hip abductor; hip flexor < hip extensor; *p* < 0.05). Among the eight dominant-leg muscles, the ankle dorsiflexor had the significantly largest rate of EMG rise (*p* < 0.05). Generally, larger perturbation intensities induced significantly shorter EMG-onset latencies (*p* < 0.05), a shorter time to peak EMG amplitude (*p* < 0.05), and a larger rate of EMG rise (*p* < 0.05).

### 3.4. MMG Signals of Eight Dominant-Leg Muscles

Figure 10 shows the mean MMG signal of the twelve subjects for each of the eight dominant-leg muscles following the unexpected balance perturbations (*n* = 12). Figure 11 plots the mean and standard error (mean ± SE, *n* = 12) values of MMG-onset latencies, time to peak MMG amplitude, as well as rate of MMG rise against the eight dominant-leg muscles.

In general, the onset of MMG signals was significantly earlier in ankle muscles than in knee or hip muscles (*p* < 0.05). Such an MMG-onset pattern (activation started from distal to proximal lower limb) was observed following all the four directions of balance perturbations. Moreover, following the balance perturbations in the frontal plane, the hip abductor was in the queue with early onset of the MMG signal. Larger perturbation intensities induced significantly shorter MMG-onset latencies (*p* < 0.05), a shorter time to peak MMG amplitude (*p* < 0.05), and a larger rate of MMG rise (*p* < 0.05).

Following sudden anterior perturbation, the ankle dorsiflexor and ankle plantarflexor had the significantly shorter MMG-onset latencies compared to the remaining muscles (*p* < 0.05). Significant differences within the agonist–antagonist muscle pairs were found in MMG-onset latency (hip flexor < hip extensor; *p* < 0.05) and in time to peak MMG amplitude (hip abductor < hip adductor; *p* < 0.05). No specific trend was observed when comparing the rate of MMG rise among muscles.

Following sudden posterior perturbation, the ankle dorsiflexor and ankle plantarflexor had the significantly shorter MMG-onset latencies compared to the other muscles (*p* < 0.05).

Following sudden medial perturbation, the ankle dorsiflexor, ankle plantarflexor, and hip abductor had significantly shorter onset latencies than the remaining dominant-leg muscles (*p* < 0.05). Significant differences within the agonist–antagonist muscle pairs were found in both the MMG-onset latency and the time to peak MMG amplitude (hip abductor < hip adductor; *p* < 0.05).

Following sudden lateral perturbation, the ankle dorsiflexor, ankle plantarflexor, and hip abductor had significantly shorter onset latencies than the remaining dominant-leg muscles (*p* < 0.05). Additionally, among the eight dominant-leg muscles, the hip abductor had the shortest time to peak MMG amplitude (*p* < 0.05).

## 4. Discussion

Via the synchronized capture of eight dominant-leg muscles’ electrical and mechanical activities (EMG and MMG signals), eight dominant-leg joint motions (angles), and whole-body postural sways (COM displacements), this study is novel in its comprehensive investigation of the timing and the speed of combined reactions in hip, knee, and ankle muscles and joints following unexpected horizontal/translational perturbations with different intensities and directions. In agreement with our hypothesis 1, this study has observed that (1) agonist muscles resisting the perturbation-induced postural sway activated more rapidly than antagonist muscles, and among the eight dominant-leg muscles, ankle muscles’ large rate of activation contributed the most to resisting the perturbations in both sagittal and frontal planes. However, our hypotheses 2 and 3 were not entirely supported, as results showed that (2) fast responses existed not only in those lower-limb joint motions and COM displacements that were passively/involuntarily induced by the perturbation but could also occur in those actively/voluntarily generated ones to counteract the perturbation; and (3) a larger perturbation intensity evoked the more rapid muscle activities but did not induce faster joint motion or postural sway.

These findings build on our knowledge of how fast the hip/knee/ankle muscles can activate and how the lower-limb joints can coordinate to maintain reactive standing balance in healthy young adults. This potentially facilitates the understanding of strategies regarding how humans cope with the varying challenges of losing balance in daily life, which could provide further evidence and guidance for the development of fall-prevention approaches in the future; the details are discussed below.

### 4.1. Faster Activation Existed in Agonist Lower-Limb Muscles, Espaecially Ankle Muscles, to Resist the Induced Postural Sway (Hypothesis 1)

The primary finding of this study was that earlier EMG onset and shorter time to peak EMG amplitude occurred in muscles that could resist the involuntary COM shift induced by the unexpected moving-platform perturbations. On top of this, this study may highlight the great contribution of ankle muscles’ large rate of activation in balance maintenance under unexpected perturbation in both the sagittal and frontal planes.

Following the unexpected anterior movement of the platform, while the COM had a firstly posterior shift relative to the BOS, the ventral muscles that could rotate the body forward (ankle dorsiflexor, knee extensor, and hip flexor) had a generally earlier onset of activation and shorter latency to reach peak activation compared to the dorsal muscles. These observations are consistent with the previously reported EMG-onset patterns (i.e., EMG-onset latency of ankle dorsiflexor < ankle plantarflexor; knee extensor < knee flexor) [12,14,45] and time to peak patterns (i.e., time to peak EMG amplitude in ankle dorsiflexor < ankle plantarflexor; knee extensor < knee flexor) [45,46]. On top of this, this study also observed that the ankle dorsiflexor had a remarkably large rate of activation after onset of perturbation (within 50 ms after onset) under all the perturbation intensities, while the knee extensor showed a large rate of activation under high perturbation intensity. This indicated the major contribution of the ventral muscles, especially the ankle dorsiflexor, in rapidly shifting the body forward to resist the unexpected anterior movement of the supporting platform. In addition, under the highest perturbation intensity, a second peak was commonly observed for the dominant-leg muscles’ EMG signals, which might be related to the subjects’ stepping action to recover postural balance. This observation is also in accordance with previous findings that a stepping strategy would be needed to increase the area of BOS and maintain balance when the ankle and/or hip strategies were insufficient [9].

Following the unexpected posterior movement of the platform, the dorsal muscle (ankle plantarflexor) showed an early onset of activation and quick reaching of peak activation to resist the induced forward COM displacement relative to BOS. This agreed with the previous findings that gastrocnemius and hamstrings had faster [10] and greater [15,46] muscle activities than the ventral muscles to prevent excessive forward postural sways. In addition, among the eight dominant-leg muscles, this study observed that the ankle plantarflexor had a notably large rising rate at the early phase of muscle activation. This corroborated the previous finding that the rate of the ankle plantarflexor’s activation played a key role in resisting the forward waist-pull balance perturbations [22] and preventing the forward tripping [20].

Following the unexpected medial movement of the platform, the hip abductor was evoked earlier and reached the peak activation earlier than the hip adductor to resist the induced lateral moving of the COM relative to the BOS. The result is consistent with previous studies that reported the significant relationship between the hip abductor and balance recovery under lateral waist-pull perturbations [22,47,48]. In addition, this study observed that the ankle dorsiflexor had a generally larger rate of activation than the other dominant-leg muscles; under the high and the highest perturbation intensities, a large rate of activation was further evoked in the hip abductor and ankle plantarflexor, as the body weight was more quickly transferred to the dominant leg. The dominant-leg distal ankle muscles have been reported in previous studies to provide the immediate joint torque to regain balance under the medial perturbation of platform, followed by the proximal hip muscles with the increasing of perturbation intensity [49,50,51]. Such kinetic responses could partially be corroborated by the further neuromuscular evidence in this study.

Following the unexpected lateral movement of the platform, the hip adductor reached peak activation earlier than the hip abductor to counteract the sudden medial moving of the COM relative to the BOS. Moreover, among the eight dominant-leg muscles, the ankle dorsiflexor had the greatest rate of activation. These findings were consistent with previous studies that indicated the essential contributions of ankle and hip muscles in controlling mediolateral postural balance [21,22,49,52]. On top of this, in the sagittal plane, this study further observed that the ventral muscles (ankle dorsiflexor, knee extensor, and hip flexor) had earlier onset and shorter time to peak activation than the dorsal muscles. These rapid activation patterns might be related to the attempt at trying to counteract the body weight unloading from the dominant leg and prevent the excessive medial postural sway induced by the unexpected lateral moving of platform [53].

To the authors’ knowledge, no previous study has investigated the activation pattern of all the major hip, knee, and ankle muscles to maintain balance over the unexpected platform movement. By examining the hip adductor/abductor, hip flexor/extensor, knee flexor/extensor, and ankle dorsiflexor/plantarflexor simultaneously, this study identified the essential importance of ankle muscles’ rapid activation in resisting horizontal perturbations for young adults. It is also expected that this study’s more comprehensive and in-depth investigation in the eight leg muscles’ activities could facilitate the future development of programs and assistive devices to improve balance and prevent falls in older adults [53,54,55,56,57,58,59,60,61,62,63,64].

### 4.2. Rapid Kinematic Responses Varied with the Perturbation Direction (Hypothesis 2)

The secondary finding of this study was that the rapid responses of lower-limb joint motions and postural sways varied in different directions of balance perturbations. Following the unexpected platform movement, lower-limb joint motions caused by the inertia of body segments were generally the first to appear and are considered as passive/involuntary, while sometimes, other active/voluntary joint motions could also appear as early as the passive/involuntary ones. Among the six directions of COM displacement, COM moving toward the opposite direction of the balance perturbation was generally the earliest because of inertia, which is consistent with the previous studies’ findings [10,58,65], while sometimes, the COM displacement in the vertical direction could also occur as early as in the horizontal direction either because of inertia or active/voluntary reactions.

Early onset of joint angles and short time to peak angles were generally observed in the passive/involuntary joint motions induced by the moving platform. Specifically, because of inertia, faster responses of ankle dorsiflexion (compared to ankle plantarflexion) [10], hip adduction (compared to hip abduction) [22,65], and hip abduction (compared to hip adduction) [22,65] were induced under the unexpected posterior, medial, and lateral perturbations, respectively. However, only under the medial perturbation was the onset of passively induced hip adduction found to be significantly earlier than the remaining seven lower-limb joint motions. For the anterior, posterior, and lateral directions, the passive/involuntary joint motion was usually accompanied by the fast responses of other active/voluntary joint motions to resist the balance perturbation.

The timing patterns of active/voluntary lower-limb joint motions were different following the four directions of unexpected balance perturbations. Specifically, under the unexpected anterior perturbation, the passively/involuntarily induced ankle plantarflexion was accompanied by the consistently early onset of hip flexion (compared to hip extension). A previous study also reported that early ankle plantarflexion was followed by the knee flexion, and hip flexion under the sudden forward movement of platform [14]. On top of it, this study observed that knee flexion (compared to knee extension) and hip flexion (compared to hip extension) reached peak angles more quickly, which further corroborates the finding in the onset of joint motions. Under the unexpected posterior perturbation, no consistently fast response within the knee or hip joint accompanied the passively/involuntarily induced ankle dorsiflexion. However, a previous study reported that young adults also had an earlier hip extension than hip flexion apart from the ankle joint motions following the posterior moving-platform perturbation [10]. This may be because the perturbation intensity induced by the posterior movement of the platform was not as large as that in the previous study [10,22]. Therefore, this study observed only the consistently later onset and peak in ankle joint motion (ankle dorsiflexion < ankle plantarflexion). The corrective reaction, i.e., ankle plantarflexion, was sufficient to recover from the induced forward loss of balance. Under the unexpected medial perturbation, all the active/voluntary joint motions appeared after the passively/involuntarily induced hip adduction. The hip abduction was corrective, as the muscle activity of the hip abductor was detected, which could work to oppose the passive hip adduction. This reaction was also supported by a previous study that found the corrective response of hip abduction momentarily following perturbations in frontal plane [22,65]. Under the unexpected lateral perturbation, the passively/involuntarily induced hip abduction was accompanied by the consistently early onset and short time to peak of hip flexion, knee extension, and ankle dorsiflexion. Similar to the medial perturbation, the hip adductor muscle worked to oppose the passive hip abduction and produce the later corrective hip abduction [22,65]. In addition, the observations of this study may suggest that when the body weight was unloaded from the dominant leg following lateral perturbations, fast joint motions in the sagittal plane were also required to maintain balance, while they were not required when the body weight was loaded on the dominant leg following medial perturbations.

The unexpected horizontal perturbation also induced the consistently faster response of COM in the vertical direction. As far as the authors know, previous studies have rarely compared the timing of postural sways in horizonal and vertical directions following a balance perturbation. Specifically, in this study, the faster upward COM displacements (compared to the downward direction) were observed following the anterior, medial, and lateral balance perturbations, while the posterior perturbation induced a faster downward COM displacement. Following the anterior perturbation, the earlier upward COM displacement (compared to downward direction) seemed to be related to the great proportion of subjects’ stepping strategies and toe-rise strategies in the study. Following the medial or the lateral perturbation, the faster responses of hip flexion, knee flexion, and ankle dorsiflexion in the unloaded leg were observed. The fast upward COM displacement could be caused by the lifting of the unloaded leg. By contrast, following the posterior perturbation, the COM reached peak displacement more quicky in the downward direction than in the upward direction. This response was considered to be passive, as the downward and the passively induced forward postural sways had similar onset latencies and time to peak. The posterior displacement quickly induced the subject’s forward inclined posture, so the COM had a firstly downward displacement and was followed by a later upward displacement to recover the upward posture. Difference in perturbation intensities may explain why the initial COM displacement in the vertical direction following posterior perturbations differed from that following anterior/medial/lateral perturbations. Posterior perturbations were set with smaller intensities than the anterior/medial/lateral perturbations in this study. The latter ones were challenging enough and induced the faster active/voluntary reactions that elevated the COM, while the former ones did not.

### 4.3. Larger Perturbation Intensity Evoked Faster Rosponse in Muscle Activation (Hypothesis 3)

The tertiary finding of this study was that larger perturbation intensity induced an earlier EMG onset, shorter time to peak EMG amplitude, and larger rate of EMG rise in the dominant-leg muscles but did not lead to faster responses in postural sways or lower-limb joint motions. This may suggest that under a larger unexpected challenge to loss of balance, the lower-limb muscles could activate earlier and more rapidly to restrict excessive joint motions and prevent excessive COM shifts out of the BOS.

The results in this study indicate that the response rate of lower-limb muscles could be related to perturbation intensity in general. EMG-onset latencies and time to peak EMG amplitude were significantly longer under the smaller perturbation intensities than under the higher ones. Previous studies have reported such a phenomenon mainly in the sagittal plane (anterior and posterior directions) [10,14,46,66]. Furthermore, this study suggests that this trend was also applicable to perturbation in the frontal plane (medial and lateral directions). In addition, this study observed a higher EMG rising rate under a larger perturbation intensity. Such effects of perturbation intensity seemed to be more prominent in the agonist muscles that could resist postural sways induced by the unexpected moving-platform perturbations (e.g., ankle dorsiflexor under the anterior perturbations; ankle muscles and hip abductor under the medial perturbations). However, the specific type of correlation between the muscle responses and perturbation intensities has remained unclear. It would be interesting to establish some models between the two factors in future studies.

A larger perturbation intensity could evoke larger responses in lower-limb joint motions and postural sways; however, it may not be able to evoke faster kinematic responses. For the postural sways, this study found that larger perturbation intensity generally induced larger COM displacement along the line of perturbation direction. An example was that both the forward and the backward peak COM displacements increased with the anterior perturbation intensity. This was understandable since the different perturbation intensities in the study were position-controlled. However, a larger intensity of perturbation did not evoke an earlier onset or shorter time to peak COM displacement. A similar amount of time was required to reach a larger postural sway under the greater perturbation intensity. This result partly agreed with a previous study that showed increasing peak COM velocity and peak COM acceleration with perturbation intensity [61]. For the lower-limb joint motions, it was found that ankle dorsiflexion, knee flexion, and hip flexion angles increased with the perturbation intensity for all the four perturbation directions. Such strategies seemed to be able to reduce the additional horizontal excursions when the challenge to loss of standing balance became larger [45]. However, the onset or the time to peak for neither the whole-body postural sways nor lower-limb joint motions were affected by the perturbation intensity. These may suggest that earlier and faster lower-limb muscle activation would be adequate for the successful maintenance of balance following a larger balance challenge. In addition, as these findings indicate that the neuromuscular reaction time (EMG-onset latency and time to EMG peak amplitude) could be modulated by the different intensities of perturbations, EMG temporal parameters could potentially be used as the training outcome or biofeedback in perturbation-based balance training [67].

### 4.4. MMG Signals Following Balance Perturbations Merit Further Study

Attempts have also been made to process and analyze the MMG signals, aiming to examine the possible delays between the electrical and mechanical muscle activities in response to unexpected balance perturbations. A new processing method was used upon optimizing the one reported in our previous study [22]. However, the MMG signals obtained in the current study may still not fully reflect the exact mechanical activities of muscles since the onset of MMG signal was not consistently later than that of EMG signal. The observed time delay between the EMG and MMG onset was not comparable with the previous studies either. The delay in the MMG signal after the EMG signal observed in previous studies ranged from 7 ms to 30 ms instead [68,69,70,71,72]. Thus, it should be noted that the following discussion of the MMG data might be affected by the current processing method of MMG signals used in this study. Further attempts are still needed to improve the MMG data processing method in future studies.

When comparing among the eight muscles, the onset of the MMG signal under all directions of perturbation was significantly earlier at distal than proximal muscles. Such onset might be caused by the perturbation induced from a moving platform supporting standing, i.e., at the distal side of subject’s body. This is further supported by the result that the value of each parameter in MMG recorded greatly depended on pulling intensity but not for muscles or pulling directions. Although MMG showed considerable reliability to detect muscle activities in static conditions [72,73], it might not be effective in dynamic conditions, as the current processing method of MMG signals can still not effectively eliminate the noise caused by the moving platform and the movement of body segment. Further studies and optimization of the set-up are required for MMG technologies to be applied to muscle-activity measurement in dynamic situations.

### 4.5. Strengths and Limitations

This study has the below strengths. Firstly, the moving-platform perturbation system was synchronized with the Vicon system and the Trigno Wireless Biofeedback System, enabling our accurate analyses of the temporal parameters of multiple signals during balance control. Secondly, the perturbation intensity (i.e., pulling displacement) was set as a percentage of the subject’s body height so that the perturbation was a consistent challenge to postural balance across different individuals. Thirdly, this study comprehensively analyzed how fast the eight major leg muscles’ activation and eight lower-limb joint motions could occur during balance control. These findings in healthy young adults can serve as the foundation and reference for not only further studies of balance-controlling mechanisms in older adults with high fall risks, but also the further development of assistive/robotic devices for targeted balance training and fall prevention.

There are some potential limitations of this study. Firstly, only a small number of subjects were recruited for this pilot study. Future studies in older adults or a specific population should justify the sample size to make the findings more representative. Secondly, during the processing of EMG signals, this study used baseline EMG value in normal standing for normalization. Therefore, it should be noted that the rate of EMG rise in this study was based on the muscle-activation level in normal standing rather than the capacity of muscle activation. Maximal voluntary contraction (MVC) tests could be carried out in the future to examine if the amplitude-normalization methods affect the finding. Lastly, the current MMG processing method might still be not mature enough to extract the exact vibrations of lower-limb muscles. Further observation and development are needed. More tests could be performed on MMG to evaluate the reliability and validity of reflecting the mechanical muscle activities following the unexpected perturbation.

## 5. Conclusions

To conclude, among the eight dominant-leg muscles, the ankle muscles’ rapid activation contributed the most to resisting the unexpectedly induced postural sways in both the sagittal and frontal planes. Fast responses of the lower-limb joint motions and the vertical postural sways were related to the perturbation direction. Under a larger perturbation intensity, muscles reacted more rapidly, while joint motions or postural sways were not necessarily faster. These findings provide new insights on the sequence or the fast responses of multiple lower-limb muscles/joints in coping with the different levels of balance challenges. The mechanisms underlying reactive standing balance are thus better understood, which may facilitate future research on developing targeted balance-training protocol and/or technology or devices for fall prevention in older adults.

## Figures and Tables

**Figure 1 bioengineering-10-00831-f001:**
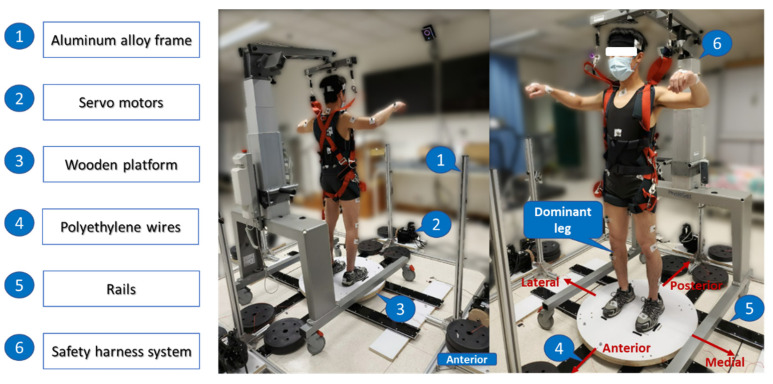
The moving-platform perturbation system with a subject.

**Figure 2 bioengineering-10-00831-f002:**
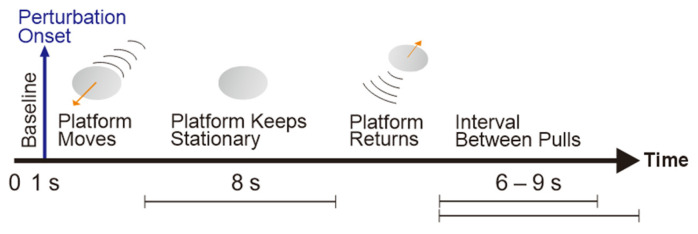
The flow of a pulling perturbation.

**Figure 3 bioengineering-10-00831-f003:**
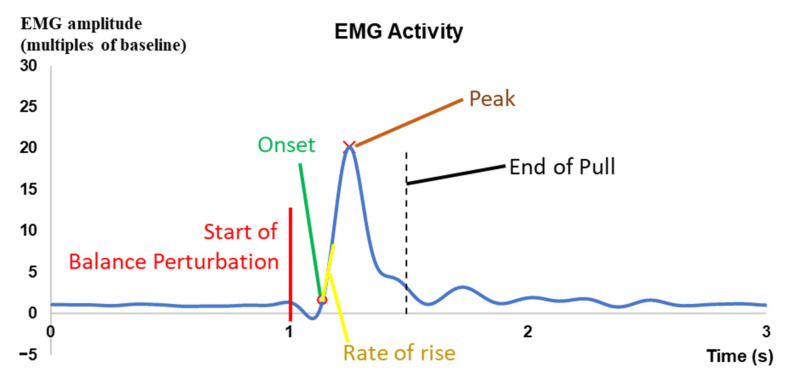
An example of EMG signal showing the examined parameters for one balance-perturbation trial.

**Figure 4 bioengineering-10-00831-f004:**
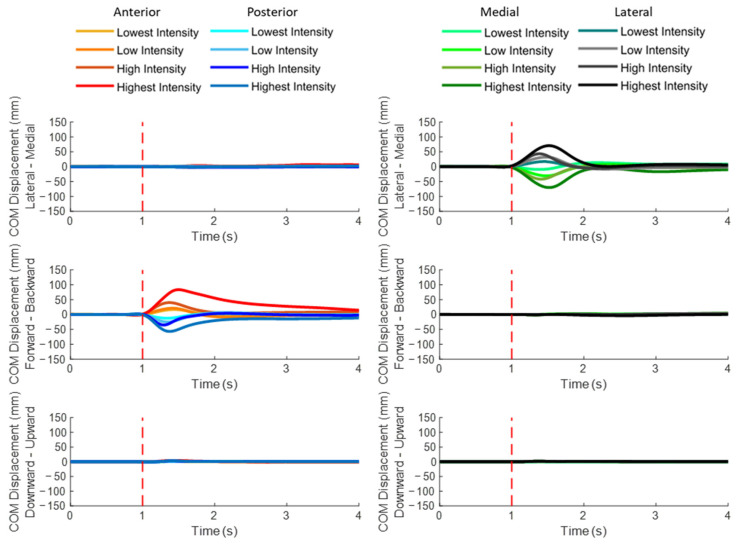
The mean whole-body COM displacements of twelve subjects following the unexpected perturbations with four directions and four intensities (*n* = 12). (Note: **COM**, center of mass; **red dotted line** specifies the start of the balance perturbation).

**Figure 5 bioengineering-10-00831-f005:**
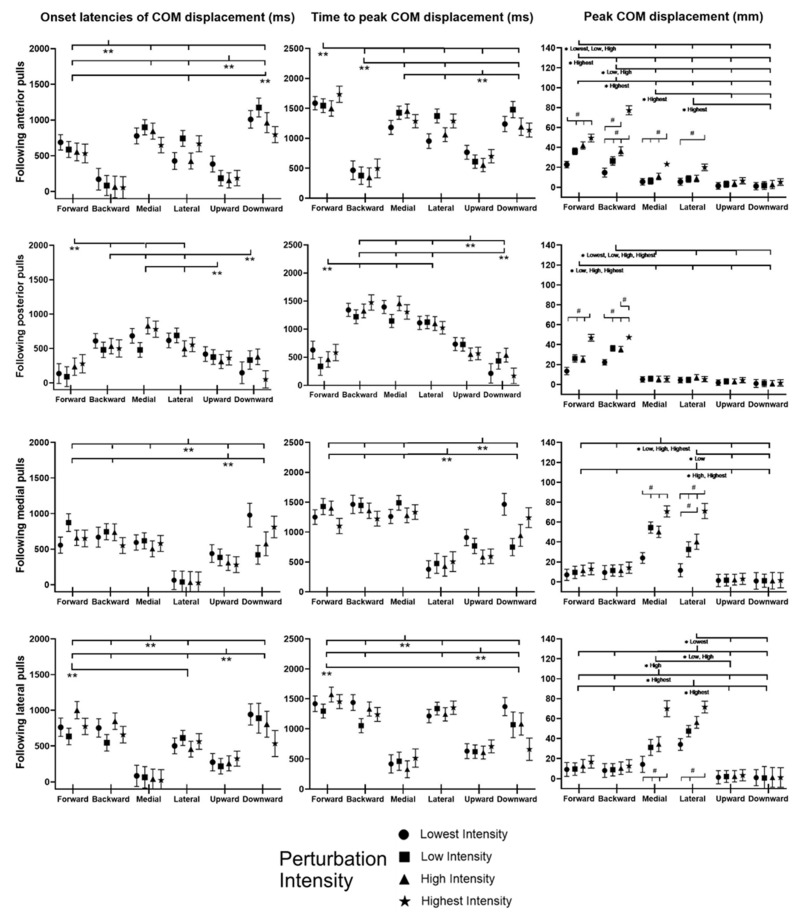
The onset latency of COM displacement, time to peak COM displacement, and peak COM displacement following unexpected horizontal perturbations (mean ± SE, *n* = 12). (Note: 
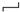
 or 
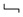
 indicates post hoc pairwise comparison; **SE**, standard error; **#** indicates significant simple main effects of intensity factor; ****** indicates significant main effects of postural sway factor; ***** indicates significant simple main effects of postural sway factor.)

**Figure 6 bioengineering-10-00831-f006:**
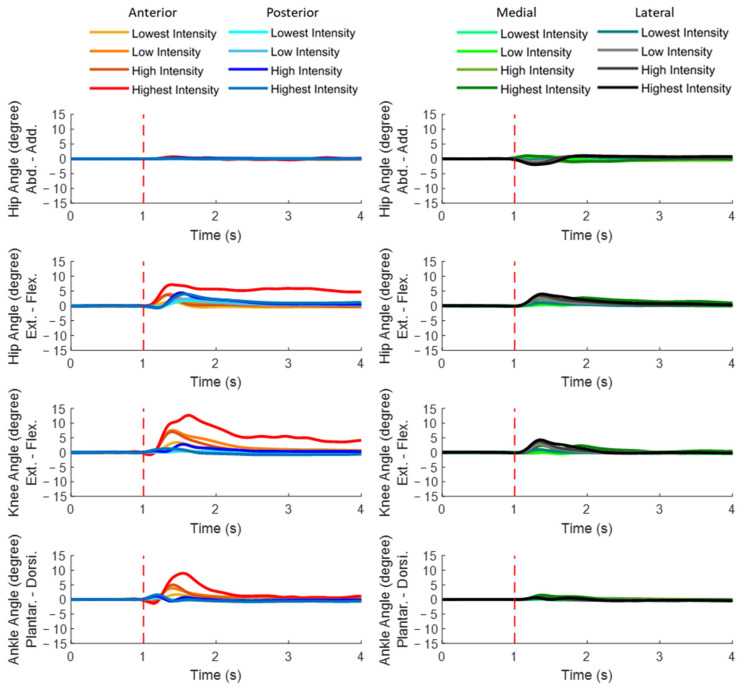
The mean dominant-leg joint angle changes of twelve subjects following the unexpected perturbations with four directions and four intensities (*n* = 12). (Note: **The red dotted line** specifies the start of the balance perturbation; **Add.**, adduction; **Abd**, abduction; **Flex.**, flexion; **Ext.**, extension; **Dorsi.**, dorsiflexion; **Plantar.**, plantarflexion.)

**Figure 7 bioengineering-10-00831-f007:**
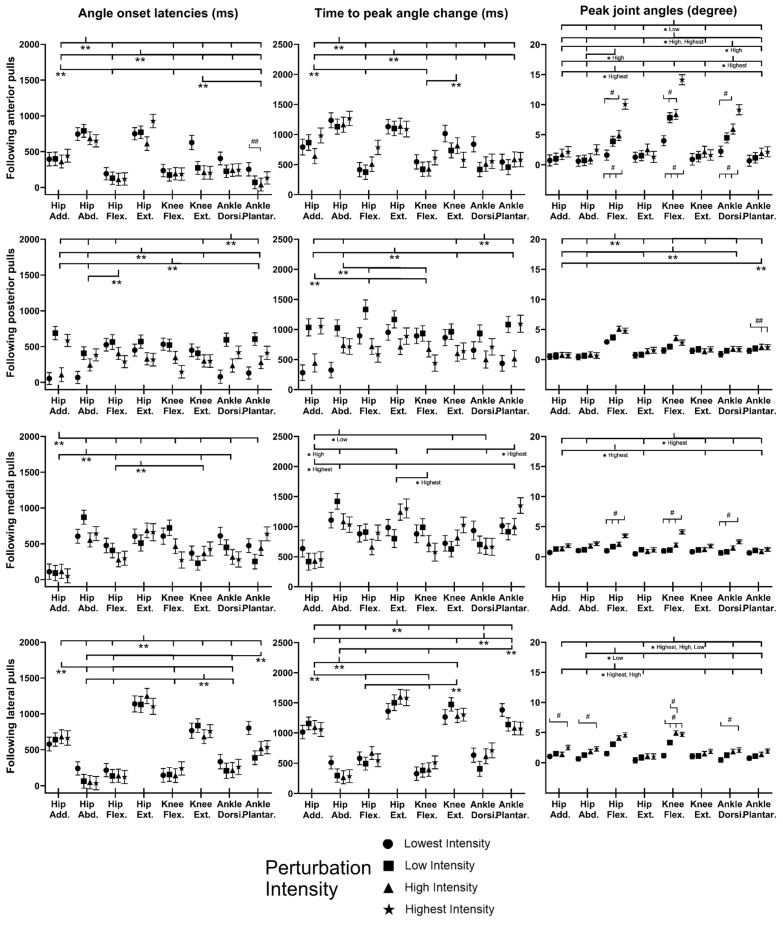
The angle-onset latencies, time to peak angle, and peak angles of eight lower-limb joint motions following unexpected horizontal perturbations (mean ± SE, *n* = 12). (Note: 
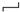
 or 
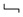
 indicates post hoc pairwise comparison; **SE**, standard error. **##** indicates significant main effects of intensity factor; **#** indicates significant simple main effects of intensity factor; ****** indicates significant main effects of joint motion factor; ***** indicates significant simple main effects of joint motion factor.)

**Figure 8 bioengineering-10-00831-f008:**
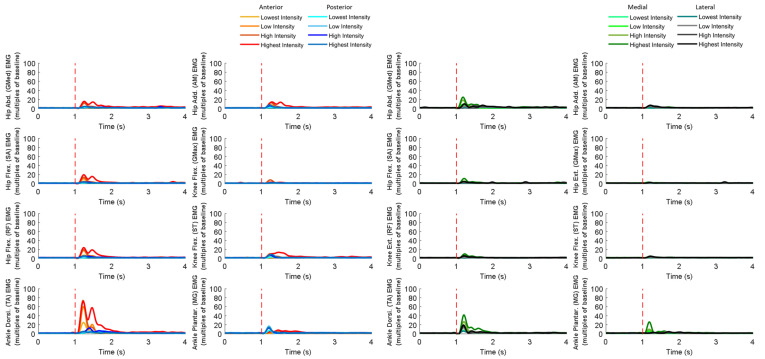
The mean EMG signal changes of twelve subjects for eight dominant-leg muscles following the unexpected perturbations with four directions and four intensities (*n* = 12). (Note: **The red dotted line** specifies the start of the balance perturbation; **EMG,** electromyography; **GMed**, gluteus medius; **AM**, adductor magus; **SA**, sartorius; **GMax**, gluteus maximus; **RF**, rectus femoris; **ST**, semitendinosus; **TA**, tibialis anterior; **MG**, gastrocnemius medialis.)

**Figure 9 bioengineering-10-00831-f009:**
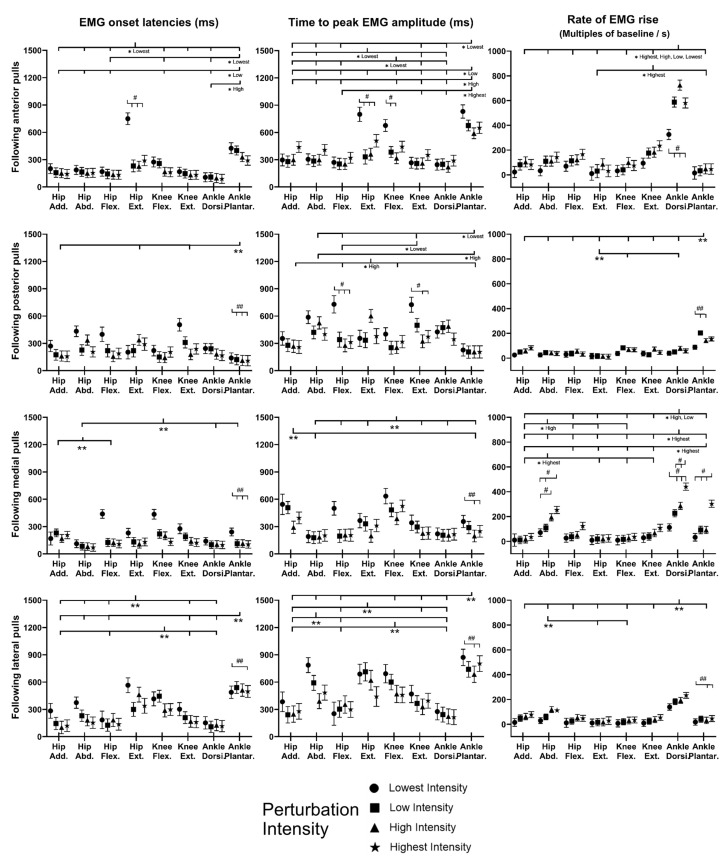
The EMG-onset latencies, time to peak EMG amplitude, and rate of EMG rise for eight dominant-leg muscles following unexpected horizontal perturbations (mean ± SE, *n* = 12). (Note: 
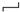
 or 
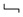
 represents post hoc pairwise comparison; **SE**, standard error; **##** indicates significant main effects of intensity factor; **#** indicates significant simple main effects of intensity factor; ****** indicates significant main effects of muscle factor; ***** indicates significant simple main effects of muscle factor.)

**Figure 10 bioengineering-10-00831-f010:**
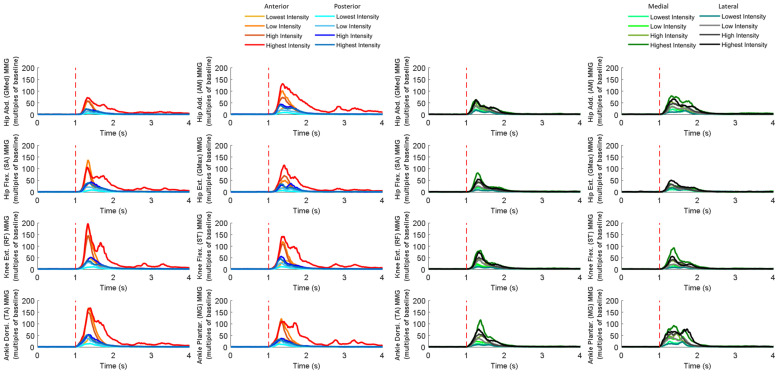
The mean MMG signal changes of twelve subjects for eight dominant-leg muscles following the unexpected perturbations with four directions and four intensities (*n* = 12). (Note: **The red dotted line** specifies the start of the balance perturbation; **MMG,** mechanomyography; **GMed**, gluteus medius; **AM**, adductor magus; **SA**, sartorius; **GMax**, gluteus maximus; **RF**, rectus femoris; **ST**, semitendinosus; **TA**, tibialis anterior; **MG**, gastrocnemius medialis.)

**Figure 11 bioengineering-10-00831-f011:**
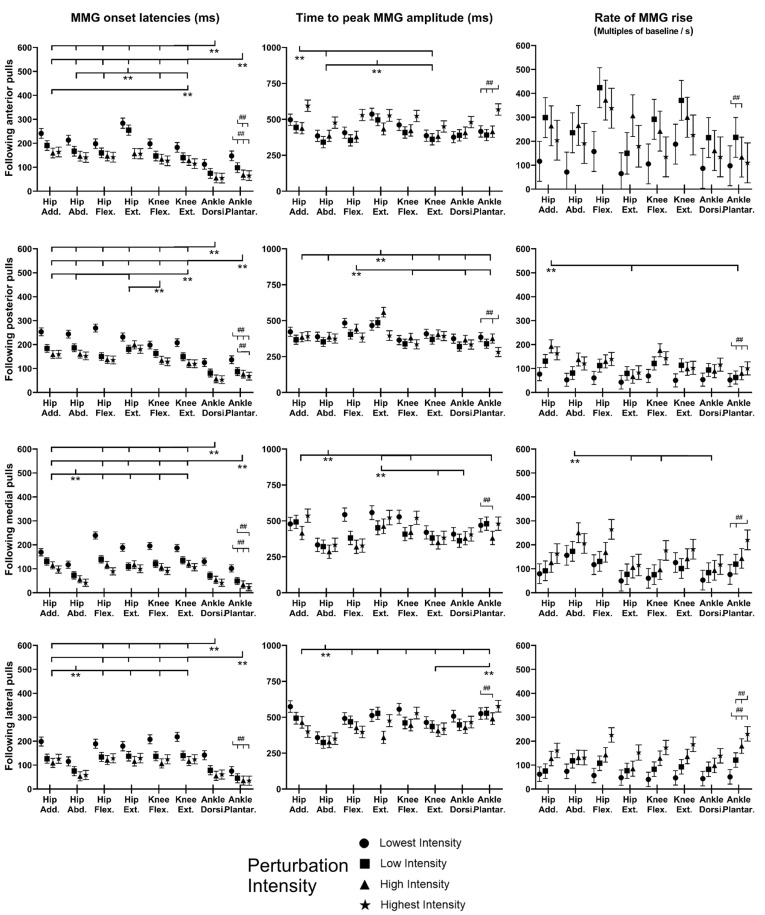
The MMG-onset latencies, time to peak MMG amplitude, and rate of MMG rise for eight dominant leg muscles following unexpected horizontal perturbations (mean ± SE, *n* = 12). (Note: 
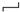
 or 
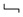
 represents post hoc pairwise comparison; **SE**, standard error; **##** indicates significant main effects of intensity factor; ****** indicates significant main effects of muscle factor.)

**Table 1 bioengineering-10-00831-t001:** Eight investigated muscles and the corresponding EMG/MMG sensor placement.

Muscle	EMG/MMG Sensor Placement
**Ankle dorsiflexor:** tibialis anterior (TA)	One-third of the way from the tip of the fibula to the tip of the medial malleolus [30];
**Ankle plantarflexor:** medial gastrocnemius (MG)	On the most prominent bulge of the muscle [30];
**Knee extensor:** rectus femoris (RF)	Halfway from the superior boarder of the patella to the ASIS [30];
**Knee flexor:** semitendinosus (ST)	Halfway from the medial epicondyle of the tibia to the ischial tuberosity [30];
**Hip flexor:** sartorius (SA)	At 8 cm distal from the ASIS along the line between the ASIS and the median of the tibial tuberosity [31];
**Hip extensor:** gluteus maximus (GMax)	Halfway from the greater trochanter to the sacral vertebrae [30];
**Hip abductor:** gluteus medius (GMed)	Halfway from the greater trochanter to the highest point of iliac crest [30];
**Hip adductor:** adductor maximus (AM)	Halfway from the medial femoral epicondyle to the pubic tubercle [32].

**Table 2 bioengineering-10-00831-t002:** Demographic data and subjective assessments (mean ± SD) of twelve subjects.

	Male (*n* = 6)	Female (*n* = 6)	Total (*n* = 12)
Age (year)	21.2 ± 1.2	21.5 ± 0.5	21.3 ± 0.9
Height (cm)	174.8 ± 5.8	166.1 ± 4.9	170.4 ± 6.9
Body Mass (kg)	59.2 ± 8.9	56.8 ± 3.6	58.0 ± 6.6
BMI (kg/m²)	19.3 ± 2.2	20.6 ± 1.0	20.0 ± 1.7
Dominant Leg	Right (*n* = 6)	Right (*n* = 6)	Right (*n* = 12)
Leg Length (cm)	88.8 ± 4.6	85.0 ± 3.2	86.9 ± 4.3
IPAQ-S (Kcal/week)	2017.3 ± 1253.3	1238.2 ± 883.6	1627.8 ± 1111.0
FES-I Short Version	10.8 ± 3.4	10.0 ± 2.8	10.4 ± 3.0
Mini-BEST Score	27.0 ± 0	27.5 ± 0.5	27.3 ± 0.5

Note: BMI, body mass index; IPAQ-S, International Physical Activity Scale—Short Version; FES-I, Fall Efficacy Scale—International; Mini-BEST, Mini-Balance Evaluation System Test.

## Data Availability

Data are contained within the article and Appendix A.

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
