# Peer review of "Muscular and Kinematic Responses to Unexpected Translational Balance Perturbation: A Pilot Study in Healthy Young Adults"

_bioengineering, 2023, doi:10.3390/bioengineering10070831_

Round 1

Reviewer 1 Report

Despite the complexity and the high number of Figures, the data relating to the results, the discussion is very well structured and systematized in a clear way, where all the problems of the present study are well understood.

The results of this investigation well support the conclusions.

Reviewer 2 Report

Thank you for the opportunity to review this paper. While the topic is of interest in its current form it will require more work before publication. There are a number of areas that require rewriting or clarification. I will comment on these areas section by section.

 ABSTRACT:

Abstract doesn't summarize the aim, methodology and conclusion clearly. Suggest it is re-written.

Title: I suggest adding "pilot study" in the title

Introduction.

The introduction is easy to read, however did not extend existing knowledge on this topic. It should include more update references regarding the association among population, falls, and health.

After the purpose statement, please provide a hypothesis for what the authors think the results will yield.

Methods
Some important information appears to be presently omitted from the methods section. Further description of the sampling procedure would be helpful for the reader. The analysis process is a bit unclear.

Some important information also appears to be presently omitted from the methods and results section. Have you tested the reliability of your data? If yes, please include the results.

Discussion
In general, the first paragraph of the discussion should at least state which hypotheses were supported. Then the authors should follow with how their results compare with similar data, and what the authors results adds to the literature (different / unique aspects of the data). Several points are made in the discussion, but it is not clear to this reviewer how results from the current study are novel or add to the literature.

The authors shortly discuss several possible explanations for the findings. The authors did not discuss what is novel about this research or what it offers in terms of health implications. The authors did not discuss how this research may be disseminated into greater practice. Moreover, the limitations and the strengths of this research were not discussed at all.

Moderate editing of English language required

Reviewer 3 Report

STRUCTURE

-       The manuscript is not correctly structured.

TITLE AND ABSTRACT

-       The Abstract is properly structured. The title is concise, specific and relevant. Type of patient is indicated.

-       The title does not include the study population.

INTRODUCTION

-       In general, the introduction is quite comprehensive, and shows a good overview of the current state of the subject.

-       the study is carried out in young adults, so the introduction must be carried out in this population. however, more than half of the introduction refers to studies carried out on older people.

MATERIAL AND METHODS

-       This section includes different relevant data of the study without differentiating it by sections, it is advisable to separate it with the following sections: design, inclusion criteria, statistical analysis, etc.

2.2. Methods

-       Line 193. What method is used to assess body composition? through biompedance, ISAK anthropometry?

-       Line 195. Used “Falls Efficacy Scale-International (FES-I) short version”. The reference is in elderly people.

-       Line 235. In this line, the researchers comment that the traction time is based on the duration of the infrared light. But could you detail how long the flash of light lasts?

RESULTS

-       The results are explained in order as they appear in the figures, well-detailed figures with footnotes and explained nomenclature.

DISCUSSION

-       Line 508. “in ankle and knee muscles were consistent with the findings of previous studies.” When talking about previous studies, what references are in the text 10,12,42 and 43?.

-       Line 540. in this line. To which side is the pull, dominant leg or non-dominant leg? “Distal ankle muscles have been reported in previous studies to provide the immediate joint torque to regain balance under the medial perturbation of platform, followed by the proximal hip muscles with the increasing of perturbation intensity”

-       Line 560. specify in which young or elderly population.

“It is also expected that the more comprehensive and in-depth investigation in eight leg muscles’ activities in this study could facilitate the future development of programs and assistive devices to improve balance and prevent falls”.

-       Line 563. In this section, in general, it would be necessary to add references to other studies that justify the results obtained.

“Rapid Kinematic Responses Varied with the Perturbation Direction.”

CONCLUSION

-       Conclusion (optional): has been included. Researchers have made a brief conclusion in this section.

REFERENCES

-       review some references such as number 7.

-       References are numbered in order of appearance in the text. They are correctly placed in square brackets [ ] and placed before punctuation.

-       The journal from which the article originates should be indicated by the journal name in of the journal abbreviated and in italics. For example, the abbreviation of reference number 25, should be indicated as J Am Geriatr Soc:

-       A bibliographic manager should be used so that the citations are well inserted, homogeneous and in accordance with the standards recommended by the journal. It is important that the year appears in bold and the journal in italics. Check the references because many of them are wrong (names, rules of the references, etc.).

Reviewer 4 Report

Dear Author,

In this study, entitled “Muscular and Kinematic Responses to Unexpected Translational Balance Perturbation in Healthy Young Adults”, the authors aims to explore how healthy young adults respond to the moving-platform induced balance perturbations with multiple directions and intensities, from the perspectives of postural sways, lower-limb joint motions, and lower-limb muscle activities.

Although the study has the potentiality of being shared with the scientific community, I believe that the manuscript would benefit from a minor revision with the attempt to better support their experimental setting.

1.     Abstract should start with a paragraph dedicated to a brief description of the background.

2.     Introduction:

The theoretical should clearly describe the scientific evidence that supports the hypothesis they have raised.

3.     Methods section:

-        More information should be provided about the participants’ characteristics.

4.     The Discussion should be enriched with the existing theory. The authors should clearly describe the scientific evidence that supports their findings. In addition, they should start with a first paragraph describing the main aims and then the main results.

Kind regards

Round 2

Reviewer 2 Report

The paper has been sufficiently revised. It can be accepted, considering it is a  pilot investigation. 

Kind regards

Dear Editors,

The paper could be accepted after moderate editing of English language.

Kind regards

Reviewer 3 Report

No further comments.